# Xinghamide A, a New Cyclic Nonapeptide Found in *Streptomyces xinghaiensis*

**DOI:** 10.3390/md21100509

**Published:** 2023-09-26

**Authors:** Soohyun Um, Jaeyoun Lee, Sung Jin Kim, Kyung A Cho, Ki Sung Kang, Seung Hyun Kim

**Affiliations:** 1College of Pharmacy, Yonsei Institute of Pharmaceutical Sciences, Yonsei University, Incheon 21983, Republic of Korea; soohyunum@yonsei.ac.kr (S.U.); jaeyoun1024@yonsei.ac.kr (J.L.); kacjjang@gmail.com (K.A.C.); 2College of Korean Medicine, Gachon University, Seongnam 13120, Republic of Korea; sungjinkim001@gmail.com (S.J.K.); kkang@gachon.ac.kr (K.S.K.)

**Keywords:** nonapeptide, halophile, *Streptomyces xinghaiensis*, anti-inflammation

## Abstract

Xinghamide A (**1**), a new nonapeptide, was discovered in *Streptomyces xinghaiensis* isolated from a halophyte, *Suaeda maritima* (L.) Dumort. Based on high-resolution mass and NMR spectroscopic data, the planar structure of **1** was established, and, in particular, the sequence of nine amino acids was determined with ROESY and HMBC NMR spectra. The absolute configurations of the α-carbon of each amino acid residue were determined with 1-fluoro-2,4-dinitrophenyl-l-and -d-leucine amide (Marfey’s reagents) and 2,3,4,6-tetra-*O*-acetyl-β-d-glucopyranosyl isothiocyanate, followed by LC-MS analysis. The anti-inflammatory activity of xinghamide A (**1**) was evaluated by inhibitory abilities against the nitric oxide (NO) secretion and cyclooxygenase-2 (COX-2) expression in lipopolysaccharide (LPS)-stimulated RAW264.7 cells.

## 1. Introduction

Within the complex realm of molecular biology, nonapeptides arise as intriguing and perplexing molecules that fulfill crucial functions in a diverse array of biologic processes. Consisting of a sequence of nine amino acids, these little yet powerful molecules have an impressive capacity to coordinate a diverse array of physiological activities, including the regulation of social behaviors, the influence on reproductive processes, and the modulation of intricate signaling networks inside organisms [1,2,3]. The appeal of nonapeptides is not just attributed to their structural simplicity but also to their significant influence on the complex mechanisms of living creatures. The nonapeptide family includes both well-known hormones such as vasopressin and oxytocin, which play crucial roles in social behaviors and emotional connections, as well as lesser-known members that exert control over physiological processes [4,5]. In addition to exerting cardiovascular effects, vasopressin plays a crucial role in regulating water balance inside the human body [6]. It has an impact on the kidneys’ process of the reabsorption of water, which helps to control hydration levels. Several nonapeptides have been shown to possess antibacterial characteristics, which play a role in the host’s defense against infections [7]. These compounds have the potential to function as natural antibiotics through their ability to break microbial membranes or interfere with bacterial activities. Certain nonapeptides play a role in the regulation of the immune system and the process of inflammation [8]. A nonapeptide called thymopentin has been studied for its ability to control immune responses, especially in people with certain immune deficiencies [9]. Corticotropin-releasing hormone (CRH), a peptide consisting of nine amino acids, plays a crucial role in the physiological mechanism of the body’s stress response [10,11]. The phenomenon elicits the secretion of stress hormones and has a role in the management of anxiety. The dysregulation of CRH signaling has been associated with the development and manifestation of anxiety disorders and mood abnormalities [12]. 

Marine organisms have undergone evolutionary processes that have resulted in the development of a wide range of bioactive substances, such as nonapeptides, which enable them to effectively adapt and thrive in their distinct watery habitats. Marine animals have developed a diverse array of bioactive compounds, such as nonapeptides, in order to adapt and flourish in their aquatic environments [13]. Marine nonapeptides play a significant ecological role, encompassing several functions, such as defense and communication [14]. Marine nonapeptides encompass a group of bioactive compounds found in marine organisms. Conopressin is a nonapeptide with structural similarities to vasopressin, which is present in the venom of cone snails belonging to the *Conus* species [15,16,17]. The venom produced by cone snails induces paralysis in their victims, facilitating their capture. The neurohypophysial hormone conopressin exerts an influence on ion channels inside the nervous system and modulates the release of neurotransmitters. An extensive study has been conducted on the utilization of this intervention for pain management and neurological investigations [18]. Heptavalinamide A and amantamides A and B are cytotoxic nonapeptides isolated from marine cyanobacteria. Heptavalinamide A showed cytotoxic activity against HeLa cells derived from cervical cancer, with an IC_50_ value of 2.8 μM [19]. Also, amantamides A and B showed moderate cytotoxicity against CCRF-CEM human T lymphoblast cells and U937 human histiocytic lymphoma cells. Amantamides A and B presented IC_50_ values of 8.3 μM and 5.6 μM, respectively, against CCRF-CEM cells, while A and B showed an IC_50_ of 6.1 μM and 7.3 μM, respectively, against U937 cells [20]. Marine-derived nonapeptides have diverse activities, including vasopressin-like activity and cytotoxicity, and could be the candidates of bioactive materials (Figure 1 and Table 1). 

In the course of our research into the potential for marine bacteria to produce novel bioactive natural compounds, we cultivated a *Streptomyces* strain named YSL1, chemically analyzed its growth, and found that it produced a nonapeptide. For the purpose of conducting a biological evaluation, we performed anti-inflammatory assays. We describe the structural determination of a non-described peptide, which we refer to as xinghamide A (**1**) (Figure 2), as well as the stereochemistry of xinghamide A (**1**) and the inhibitory effects of xinghamide A (**1**) against NO secretion and COX-2 expression.

## 2. Results and Discussion

A batch of nine marine bacteria samples was obtained from the Taepyeong saltern located in Shinan, South Korea. One of the strains examined in this study was strain No. YSL1, which was classified within the *Streptomyces* group. The tentative classification of the producing organism as a *Streptomyces* species was based on its colonial and microscopic morphological features. The attribution was further substantiated by the utilization of the 785F /907R primer pair for 16S rRNA gene sequencing. The subsequent investigation centered on the residual biomass of *Streptomyces* sp. YSL1. The strain was cultivated using both liquid and solid culture methods, employing a variety of bacterial culture media. A tiny amount of each sample was then subjected to extraction using ethyl acetate and methanol, employing sonication. Liquid chromatography—mass spectrometry (LC-MS/MS) with data-dependent fragmentation and scanning was used to analyze the organic extracts. This analytical approach facilitated the construction of two molecular networks using the Global Natural Product Social Molecular Networking (GNPS) platform [24]. The GNPS analysis indicated the existence of secondary metabolites that were potentially annotated without the utilization of analogue searching. One particular molecular family exhibited distinct characteristics compared to the others detected only in solid culture (Appendix A). This distinction arose from the absence of library matches for the related MS/MS spectra in both native and analog search modes. 

In particular, the mass peak at *m*/*z* 988.5087 was later found to be the sodium adduct ion [M + Na]^+^. By using a mixture of ethyl acetate and methanol as the solvent, it was possible to obtain xinghamide A (**1**) in the form of white powder. The high-resolution electrospray ionization mass spectrometry (HRESIMS) spectrum showed the presence of the sodium adduct ion [M + Na]^+^ with a mass-to-charge ratio (*m*/*z*) of 988.5087 (calculated for C_48_H_71_N_9_O_12_Na, 988.5125). The Fourier-transform infrared (FT-IR) spectrum exhibited prominent peaks at wavenumbers of 3309, 2944, 2833, 1654, and 1451 cm^-1^, indicating the existence of amide functional groups. 

The ^1^H nuclear magnetic resonance (NMR) spectrum obtained in deuterated dimethyl sulfoxide (DMSO-*d*_6_) exhibited six distinct NH signals within the chemical shift range of δ_H_ 8.17–7.13. The ^13^C NMR spectrum exhibited nine distinct carbonyl signals ranging from δ_C_ 172.2 to 168.6, as well as nine alpha carbon resonances within the δc 59.9–54.7 area (Appendix A and Table 2).

The HSQC spectrum of displayed the presence of thirteen methine carbons (δc 66.9–30.1), twelve methylene carbons (δc 47.3–23.8), and seven methyl carbons (δc 19.2–18.1) present. The available data indicate that **1** is a peptide composed of nine amino acids, including phenylalanine (Phe), threonine (Thr), glutamic acid (Glu), three valines (Val1–3), and three prolines (Pro1–3) residues facilitated by the analysis of the ^1^H-^1^H COSY spectrum in conjunction with the HSQC and HMBC spectra. The amino acid residues were subjected to additional structural assignment and sequencing by using the data of HMBC and ROESY experiments. First, the HMBC correlations from 2-NH (δ_H_ 7.13) of Phe and H-11 of Pro-1 (δ_H_ 4.17) to C-10 (δ_C_ 169.2) confirmed the connectivity of Phe to Pro-1. The ^1^H-^13^C long-range correlation from H-16 (δ_H_ 4.18) to C-15 (δ_C_ 170.6) and the ROESY correlation between H-14a (δ_H_ 3.49) and H-16 established the sequence of Pro-1 to Val-1. The sequence of Val-1 to Val-2 was determined by establishing the HMBC correlations from the NH group of Val-1 (δ_H_ 7.79) and H-21 (δ_H_ 4.13) of Val-2 to C-20 (δ_C_ 170.8). The connectivity between Val-2 and Pro-2 was determined through the HMBC long-range coupling observed from the amide proton (δ_H_ 7.93) of Val-2 to C-25 (δ_C_ 171.4) of Pro-2 and the observed ROESY connection between the α-proton at position H-26 (δ_H_ 4.41) of Pro-2 and the proton of the amide group in Val-2. The amino acid adjacent to Pro-2 was identified as Val-3 based on the ^1^H-^13^C long-range correlation from the α-proton at H-31 (δ_H_ 4.28) of Val-3 to C-30 (δ_C_ 169.4) and ROESY correlation between H-29b (δ_H_ 3.64) of Pro-2 and H-31 of Val-3. Glu was connected to Val-3 by the ROESY correlation between H-32 (δ_H_ 1.92) and H-37a (δ_H_ 2.21) and the HMBC correlation from 31-NH (δ_H_ 7.81) to the carbonyl carbon C-35 (δ_C_ 170.5) of Glu. The connection between Glu and Pro-3 was established through the analysis of the ROESY correlation between H-38b of Glu and H-41 (δ_H_ 4.43) of Pro-3. The connection between Pro-3 and Thr was determined by the use of ROESY analysis revealing a coupling between H-44a (δ_H_ 3.72) of Pro-3 and the α-proton (δ_H_ 4.38) of Thr. Based on the HMBC correlations from 46-NH (δ_H_ 8.17) of Thr and the α-proton (δ_H_ 3.08) of Phe to the carbonyl carbon (δ_C_ 172.2) of Phe, the final peptide residue, Thr, was located adjacent to Phe deducing the macrocyclic ring connecting Phe to Thr. Additionally, the confirmation of the amino acid residue sequence was substantiated by the ESIMS/MS experiment (Appendix A). Based on the chemical formula’s indication of 18 unsaturated number, the aforementioned data suggests the existence of a cyclic nonapeptide (Figure 3 and Figure 4).

Advanced Marfey’s method was used to determine the absolute configurations of the amino acid residues in **1**. The compound was exposed to acid hydrolysis using a 6 N hydrochloric acid solution at a temperature of 110 °C overnight. The hydrolysate underwent derivatization using Marfey’s reagents, namely 1-fluoro-2,4-dinitrophenyl-l-and -d-leucine amide (l- and d-FDLA). The absolute configurations of the α-carbon of each amino acid residue were determined by analyzing the derivatives using LC-MS [25]. It was found that all of the amino acids have an l-configuration (Appendix A). The determination of the stereochemistry of the extra stereogenic center in the threonine was accomplished through a comparative analysis of the retention time of its derivative, 2,3,4,6-tetra-*O*-acetyl-β-d-glucopyranosyl isothiocyanate (GITC), with the retention times of GITC derivatives obtained from authenticated standards of *allo*-l-threonine and l-threonine [26].

To analyze the anti-inflammatory activity of xinghamide A (**1**), we measured cell viability using RAW264.7 cells. RAW264.7 cells were treated with xinghamide A (**1**) at concentrations of 6.3 to 200 μM for 2 hr, followed by a 22 h treatment with 500 ng/mL of LPS. As shown in Figure 5A, a slight decreasing compared to the LPS group at the concentration of 200 μM. On the other hand, in the next study, xinghamide A (**1**) showed a similar cell viability to the LPS group at concentrations of 6.3, 12.5, 25, 50, and 100 μM. Therefore, the following study was conducted using concentrations conducted using concentrations below 200 μM.

Next, we investigated the inhibitory effect of xinghamide A (**1**) on NO secretion in LPS-stimulated RAW264.7 cells. As shown in Figure 5B, significant reductions were confirmed at xinghamide A (**1**) concentrations of 200 and 100 μM (each NO secretion, 2.5 and 12.6 μM), compared to the positive control group (NO secretion, 18.0 μM). Therefore, we analyzed COX-2 expression to elucidate the molecular mechanisms underlying its anti-inflammatory activity.

Then, we investigated whether xinghamide A (**1**) inhibits LPS-induced COX-2 protein expression. RAW264.7 cells were treated with xinghamide A (**1**) at concentrations of 10, 50, and 100 µM for 2 h, followed by a 22 h treatment with LPS. As describe in Figure 6A, the results revealed a concentration-dependent reduction in COX-2 protein expression following xinghamide A (**1**) treatment. Figure 6B visually presents the intensity of immunoblot bands. These results indicate that xinghamide A (**1**) treatment significantly reduces COX-2 expression.

NO plays an important role in the inflammatory response. It is generated by a group of enzymes called nitric oxide synthases (NOS) [27]. NO is known to have beneficial roles in anticancer activity, antiviral effects, and anti-inflammatory processes [28]. However, the excessive production of NO by macrophages is associated with inflammatory and autoimmune diseases [29]. Therefore, controlling the excessive production of NO has been highlighted as a therapeutic strategy for chronic inflammatory conditions. In this study, we demonstrated that treatment with xinghamide A (**1**) in LPS-stimulated RAW264.7 cells has an inhibitory effect on NO secretion. Lipopolysaccharides (LPS) bind to a specific receptor called Toll-like receptor 4 (TLR4), inducing various inflammatory responses in various cell types, particularly macrophages [30,31]. During the inflammatory process, TLR4 is overexpressed and activated by LPS, leading to an increase in COX-2 expression and production [32]. Considering the significant role of COX-2 in the signaling pathway activated by LPS and its importance in the production of inflammation, inhibiting COX-2 expression holds great significance in regulating inflammatory responses. Our results demonstrate that xinghamide A (**1**) treatment effectively inhibits COX-2 expression in LPS-stimulated RAW264.7 cells, suggesting that xinghamide A (**1**) could be considered as a novel therapeutic strategy for inflammatory disease treatment.

## 3. Experimental Section

### 3.1. General Experimental Procedures

Optical rotation was measured on an Optronic polarimeter P3000 (KRÜSS GmbH, Hamburg, Germany). UV spectra were obtained with a Cary 100 UV-VIS spectrophotometer (Varian, Palo Alto, CA, USA) with a 1 cm micro quartz cuvette, and IR spectra were with Cary 630 FTIR (Agilent Technologies, Santa Clara, CA, USA). NMR spectra were recorded with DMSO-*d*_6_ on Bruker Avance III 850 HD instruments (Bruker BioSpin, Billerica, MA, USA), using NMR solvent peaks (DMSO: δ_C_ 39.5; δ_H_ 2.50) as internal chemical shift references. Proton and carbon NMR spectra were measured at 850 and 212.5 MHz, respectively. High-resolution mass spectrometric data were acquired on Agilent 6530 iFunnel quadrupole-time of flight mass spectrometer coupled with Agilent 1290 UHPLC system at 30 °C. Compound was purified using Agilent 1100 series capillary LC system linked with a Waters micromass ZQ mass spectrometer (Waters Corp., Milford, MA, USA).

### 3.2. Collection and Identification of Bacterial Material

Specimens of a halophyte, *Suaeda maritima* (L.) Dumort, were collected from Shinan saltern (34° 59′ 53.0″ N 126° 10′ 19.4″ E), Republic of Korea in October 2021. The roots of *S*. *maritima* were disinfected by soaking in 5% sodium hypochlorite for 5 minutes and wiping with 70% aqueous ethanol. The sterilized root parts were flaked and put onto chitin, Czapek-Dox, and A1 solid media for 14 days to isolate bacteria of *S*. *maritima*. Each medium contained 33 g sea salt per 1 L of sterilized water to isolate halophile bacteria; chitin medium (6 g chitin, 0.75 g K_2_HPO_4_, 0.5 g MgSO_4_·7H_2_O, and 3.5 g K_2_HPO_4_, 10 mg FeSO_4_·7H_2_O, 10 mg MnCl_2_·4H_2_O, 10 mg ZnSO_4_·7H_2_O, 100 mg cycloheximide, 33 g sea salt, and 36 g agar per 1 L of sterilized water); Czapek-Dox medium (30 g sucrose, 2 g NaNO_3_, 1 g K_2_HPO_4_, 0.5 g MgCl_2_, 0.5 g KCl, 0.01 g FeCl_2_, 100 mg cycloheximide, 33 g sea salt, and 18 g agar per 1 L of sterilized water); A1 medium (10 g starch, 4 g yeast extract, 2 g peptone, 100 mg cycloheximide, 33 g sea salt, and 18 g agar per 1 L of sterilized water). Colonies from three media were inoculated repeatedly to fresh LB-PDB mixed solid media to obtain single strains, and one of the bacterial strains was identified as *Streptomyces xinghaiensis* based on the 16S rRNA gene sequence. The 16S rRNA sequence was submitted to the GenBank database and assigned the accession number OM992324.

### 3.3. Fermentation, Extraction, and Isolation

After culturing *S*. *xinghaiensis* in LB-PDB mixed solid media (12 g of LB, 12 g PDB, 1 g TSB, 33 g sea salt, and 18 g agar per 1 L of sterilized water), the colonies were transferred to 250 mL Erlenmeyer flask containing 100 mL of LB-PDB mixed medium. Then, 10 ml broths of preculture were spread on LB-PDB mixed solid media (150 mm Petri dishes) and cultivated for 14 days at 28 ℃. The agar media covered with *S*. *xinghaiensis* were cut into 1 cm × 1 cm pieces and extracted with ethyl acetate followed by methanol. The dried methanol extract (24 g) was loaded onto a prepacked SPE-C18 column (S*Pure, Singapore) and stepwise eluted (20, 40, 60, 80, and 100% aqueous methanol). The targeted compound was eluted with 40% aqueous methanol (5.8 g), and the fraction was further purified with semipreparative LC-MS instruments. The HPLC conditions consisted of 28% aqueous acetonitrile with 0.1% formic isocratic system as mobile phase using a YMC J’ sphere ODS H80 column (250 × 20 mm, 4 μm). Xinghamide A (**1**) was eluted at a retention time of 23 min and 2.7 mg of pure compound was obtained after removing the solvent.

*Xinghamide A* (**1**):

white powder; [α]_D_^25^ + 3.8 (c 0.1, MeOH); UV (MeOH); λ_max_ (log ε) 221 (3.50), 201 (2.33) nm; IR (ATR) ν_max_ 3309, 2944, 2833, 1654, 1451, 1259, 1024 cm^-1^; ^1^H and ^13^C NMR (DMSO-*d*_6_) see Table 2; HRESIMS *m*/*z* 988.5087 [M + Na]^+^ (calcd. for C_48_H_71_N_9_O_12_Na, *m*/*z* 988.5125).

### 3.4. Analyses of Secondary Metabolites and Molecular Networking

For comparison of secondary metabolites according to the culture conditions and extract solvents of *S*. *xinghaiensis*, liquid cultures and solid cultures were analyzed individually. *S*. *xinghaiensis* cultivated in LB-PDB mixed liquid media for 14 days was extracted with ethyl acetate/water layer separation using a separatory funnel, and the ethyl acetate layer was concentrated in vacuo. The LB-PDB mixed agar plate was extracted with ethyl acetate, followed by methanol, and their corresponding extracts were dried *in vacuo*. Each dried extract was dissolved in methanol at a concentration of 250 μg/ml and analyzed using liquid chromatography—mass spectrometry (LC-MS), using YMC-Triart C18 column (150 × 2.0 mm, 5 μm) (YMC Korea, Seungnam, Korea). The MS experiment was performed under the following conditions: drying gas temperature 300 °C, drying gas flow rate 8 L/min, sheath gas temperature 350 °C, sheath gas flow rate 11 L/min, and capillary voltage + 3.5 kV with positive mode (pressure of nebulizer 35 psig, capillary voltage 3500 V, fragmentor 175 V, skimmer 65 V, and OCT 1 RF Vpp 750 V).

HRMS^2^-based GNPS (Global Natural Product Social Molecular Networking) analysis was conducted on the MS/MS data from the ethyl acetate crude extracts of liquid and of solid cultures and from the methanol crude extract of the solid culture after 14 days of cultivation of *S*. *xinghaiensis* YSL1. The data were converted to the .mzML format with MS-Convert, and the converted files were used to generate an MS/MS molecular network using the GNPS web-server. The precursor ion mass tolerance was set to 2.0 Da and the product ion tolerance was set to 0.05 Da. The molecular networks were generated using a minimum of 6 matched peaks, a cosine score of 0.7, and a minimum cluster size of 2 to 3. After analyzing, data were visualized using Cytoscape 3.10.0 software [33].

### 3.5. Analysis of the Configuration of Amino Acids in Xinghamide A *(**1**)*

The absolute configuration of amino acids in xinghamide A (**1**) was determined with advanced Marfey’s reaction. A total of 1 mg of xinghamide A (**1**) was hydrolyzed with 0.5 mL of 6 N aqueous HCl solution at 115 °C with stirring for 24 hr, and then the reaction vials were cooled in an ice bath for 3 min. The reaction mixtures were dissolved in water and dried in vacuo. To remove residual HCl, the process was repeated three times and lyophilized for 24 hr. The acid hydrolysates of xinghamide A (**1**) containing the free amino acids were divided into two vials, and each portion was dissolved in 100 μL of 1 N NaHCO_3_. To each solution, either 50 μL of 10 mg/mL l-FDLA (1-fluoro-2,4-dinitrophenyl-5-l-leucine amide) or d-FDLA (1-fluoro-2,4-dinitrophenyl-5-d-leucine amide) in acetone was added. The reaction mixtures were heated at 80 °C for 3 min and quenched by adding 50 μL aliquot of 2 N HCl, followed by the addition of 300 μL of aqueous 50% acetonitrile. The additional stereocenter at β-carbon of threonine was defined by comparing the retention time of its GITC (2,3,4,6-tetra-*O*-acetyl-β-d-glucopyranosyl isothiocyanate) derivative with the retention times of the GITC derivatives of standards of l-Thr, d-Thr, allo-l-Thr, and allo-d-Thr. The hydrolysate of xinghamide A (**1**) was dissolved in H_2_O at a concentration of 1 mg/mL in a 5 mL vial. Then, 100 μL of 6% triethylamine and 100 μL of 1% GITC in acetone were added to the vial. The reaction was conducted at room temperature for 15 min, and the reaction mixture was diluted with 100 μL of 5% acetic acid. Each 10 μL aliquot reaction mixture was analyzed by LC-QTOF-MS under a gradient solvent system (30 to 50% aqueous acetonitrile containing 0.1% formic acid over 50 min, 0.4 mL/min flow rate, Phenomenex Luna reversed-phase C18 column 100 × 4.6 mm, 5 μm). The retention times of the l- and d-FDLA-derivatized hydrolysates were 19.2 min and 28.7 min for Thr, 23.5 min and 25.4 min for Glu, 27.5 min and 33.9 min for Pro, 36.1 min and 52.1 min for Val, 45.2 min and 55.7 min for Phe, respectively. Thus, all the amino acids in xinghamide A (**1**) were determined to be l-amino acids. Also, the GITC derivative in xinghamide A (**1**) was eluted with a retention time at 6.8 min with each authentic standard l-Thr (6.8 min), d-Thr (7.2 min), allo- l-Thr (6.4 min), and allo- d-Thr (6.5 min). Thus, Thr in xinghamide A (1) was determined to be l-Thr.

### 3.6. Biological Assays

#### 3.6.1. Cell Culture

RAW264.7 cells (Korean Cell Line Bank, Seoul, Korea) were cultured in DMEM medium (Corning Inc., Corning, NY, USA) supplemented with 10% FBS and 1% penicillin-streptomycin. The cells were maintained in 100 mm culture dishes and subculture based on attaining 80% confluence every 2 days. The culture conditions included 37 °C temperature and 5% CO_2_ with humidity.

#### 3.6.2. Cell Viability and NO Secretion Assay

RAW264.7 cells were plated at a density of 0.5×10^5^ cells/well in a 96-well plate. On the following day, xinghamide A (**1**) was diluted in DMEM medium to final concentrations of 6.5, 12.5, 25, 50, 100, and 200 µM for 2 hr. After, the cells were stimulated to Lipopolysaccharides (LPS) at a concentration of 500 ng/mL for 22 h. Subsequently, the culture medium was harvested and replaced with medium containing EZ-Cytox (DogenBio, Seoul, Korea) to cell viability measurement. The collected culture supernatant was utilized for NO secretion analysis through the Griess assay, with nitric oxide concentration determined by measuring optical density at 550 nm using a sodium nitrate standard curve.

#### 3.6.3. COX-2 Expression Analysis

RAW264.7 cells were seeded at 1 × 10^6^ cells/well density in 6-well plates. After 24 h, cells were pre-treated with xinghamide A (**1**) at concentrations of 10, 50, and 100 µM, followed by stimulation with LPS (500 ng/mL) for 22 h. Cells were washed thrice with cold DPBS, lysed using radioimmunoprecipitation assay buffer (RIPA buffer) supplemented with dithiothreitol (FUJIFILM Wako, Osaka, Japan) and complete™ Mini Protease Inhibitor Cocktail (Roche Diagnostics Corp., Indianapolis, Indiana, USA). Lysates were centrifuged at 13,000 rpm for 20 min at 4 °C, and the supernatant was collected. Total protein was quantified at a concentration of 1 μg/μL. SDS-PAGE was utilized to separate 10 μg of protein per well, and then transferred to a polyvinylidene fluoride (PVDF) membrane. The membrane was blocked with 5% skim milk overnight at 4 °C. Subsequently, the PVDF membrane was incubated at 4 °C for 2 h with primary antibodies against COX-2 and β-actin (Cell Signaling, Danvers, MA, USA), diluted in TBS-T containing 0.1% Tween-20. Following three 15 min washes with TBS-T, the membrane was incubated for 1 h at room temperature with secondary antibodies conjugated with horseradish peroxidase (HRP), specific to the primary antibodies. After another three TBS-T washes, protein detection was carried out using Super Signal® West Femto Substrate (Thermo Fisher, Waltham, MA, USA) and the Fusion Solo Chemiluminescent System (Vilber Lourmat, Eberhardzell, Germany).

#### 3.6.4. Statistical Analysis

The graphs were expressed using mean ± standard deviation values obtained from duplicated analyses. Data analysis involved utilizing GraphPad Prism 8 software (GraphPad, Inc., La Jolla, CA, USA), followed by performing one-way analysis of variance after Tukey’s post hoc test.

## Figures and Tables

**Figure 1 marinedrugs-21-00509-f001:**
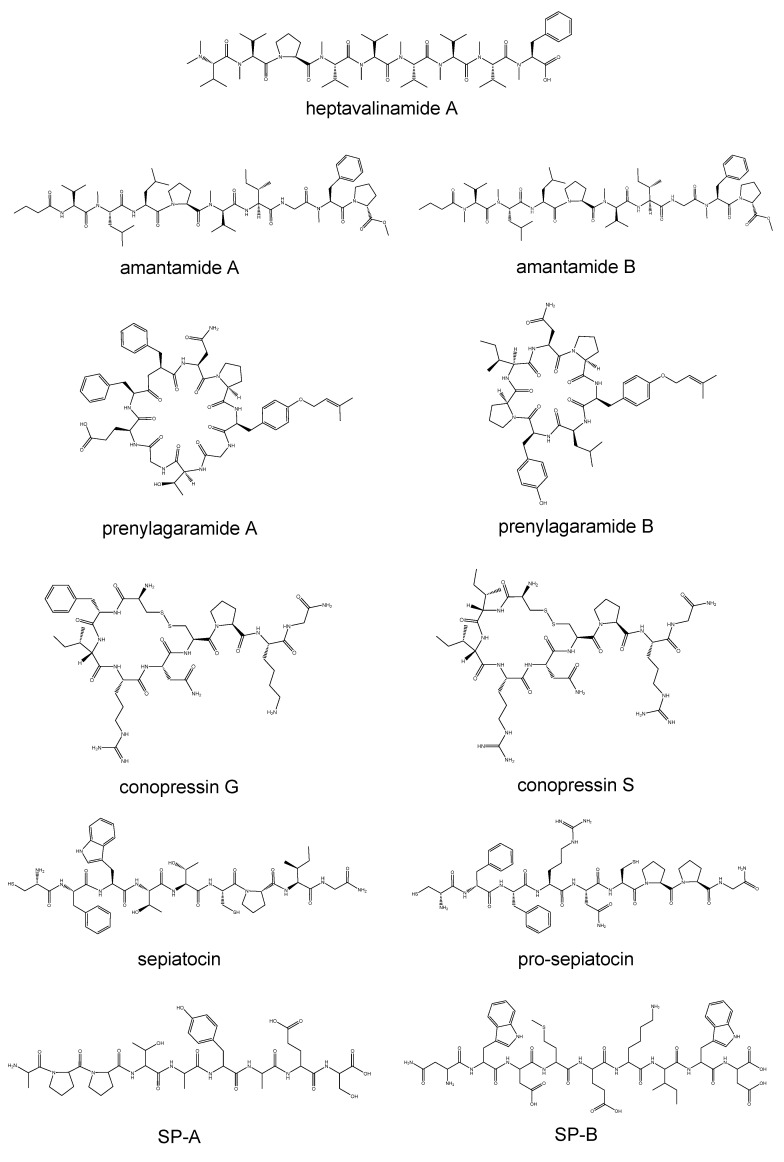
Structures of nonapeptides isolated from marine organisms.

**Figure 2 marinedrugs-21-00509-f002:**
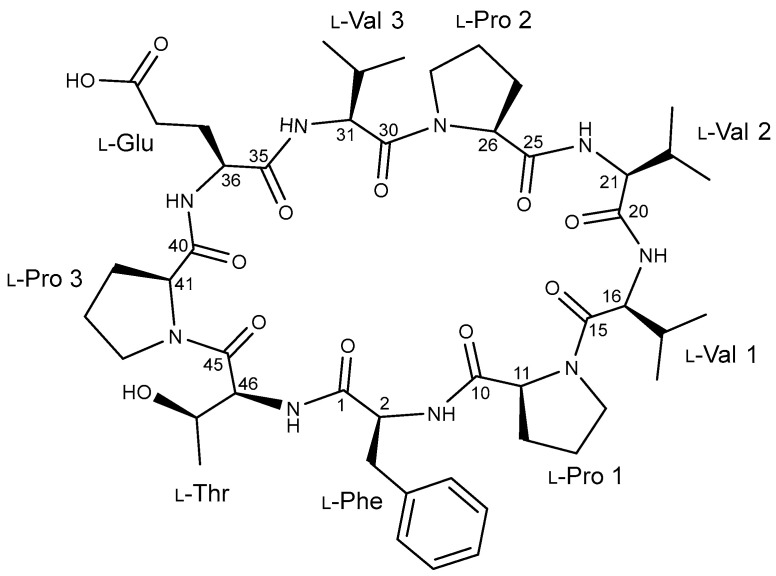
Chemical structure of xinghamide A (**1**).

**Figure 3 marinedrugs-21-00509-f003:**
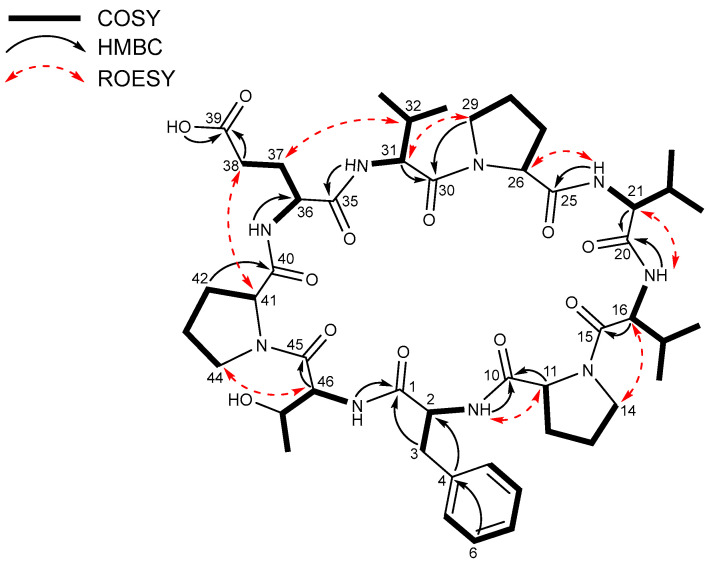
Key COSY, HMBC, and ROESY correlations in xinghamide A (**1**).

**Figure 4 marinedrugs-21-00509-f004:**
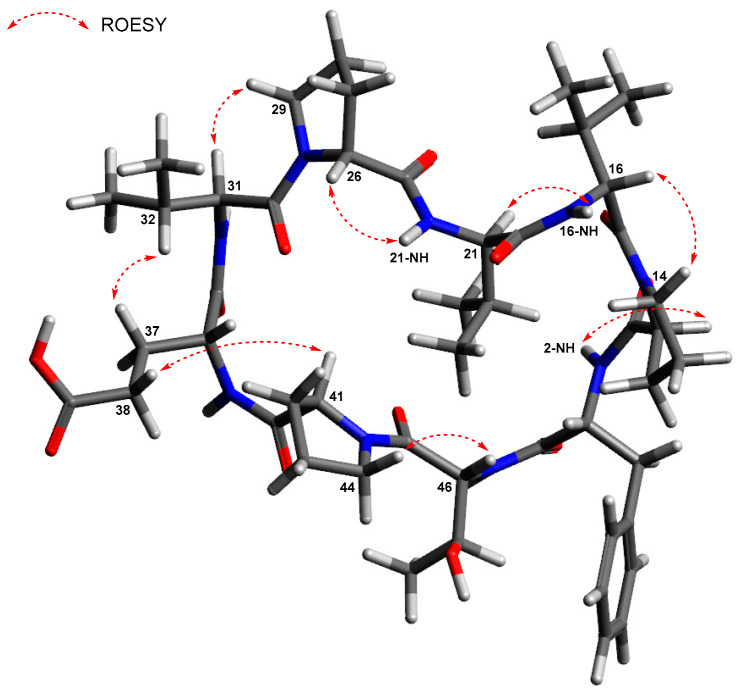
The energy-minimized structure and the key ROESY correlations in xinghamide A (**1**).

**Figure 5 marinedrugs-21-00509-f005:**
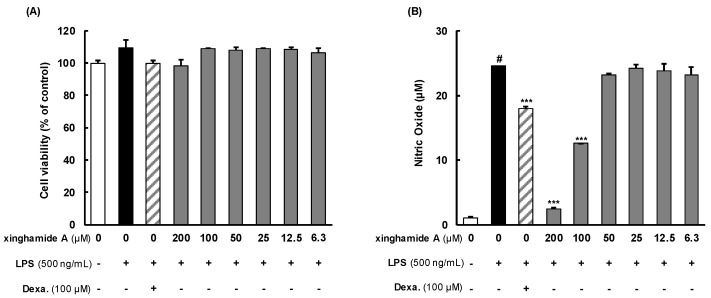
Effects of xinghamide A (**1**) on cell viability and nitric oxide secretion in lipopolysaccharide (LPS)-induced RAW264.7 cell. (-): DMEM treated groups, (+): LPS-treated group or Dexamethasone (Dexa.)-treated group. (**A**) RAW264.7 cells (0.5 × 10^5^/well) pre-treated with xinghamide A at a concentration of 6.3, 12.5, 25, 50, 100, or 200 μM for 2 h and LPS (500 ng/mL) for 22 h. Cell viability was determined by using EZ-cytox reagent as described in the method sections. (**B**) RAW264.7 cells (0.5 × 10^5^/well) pre-treated with xinghamide A at a indicated concentrations for 2 h and LPS for 22 h. Cell supernatants were harvested, and nitric oxide production was analyzed by Griess reagents. Dexa was used as a positive control. All data were calculated mean ± SD of duplicate. ^#^
*p* < 0.0001 compared to control and *** *p* < 0.0001 compared to LPS-treated groups.

**Figure 6 marinedrugs-21-00509-f006:**
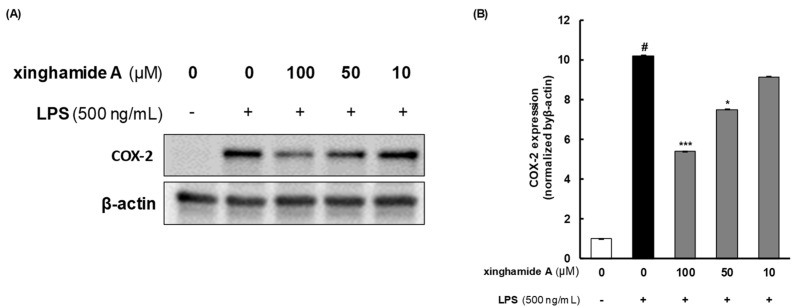
Effects of xinghamide A (**1**) on COX-2 expression in LPS-induced RAW264.7. (**A**) RAW264.7 cells (1 × 10^6^/well) pre-treated with xinghamide A at a concentration of 10, 50, or 100 μM for 2 h and LPS (500 ng/mL) for 22 hr. COX-2 expression was evaluated using immuno-blotting with the specific COX-2 antibody. (**B**) Bar graph of the intensity of western blot bands visualized using Image J software (version (x64)1.8.0). All data were calculated mean ± SD of duplicate. ^#^
*p* < 0.0001 compared to control and *** *p* < 0.0001 and * *p* < 0.05 compared to LPS-treated groups.

**Table 1 marinedrugs-21-00509-t001:** Nonapeptides from marine organisms and their biological activities.

Origins	Species	Bioactive Compounds	Activities	References
Marine Microbial				
marine cyanobacteria	*Symploca* sp.	heptavalinamide A	cytotoxic	Suo, R et. al. [19]
	*Oscillatoria* sp.	amantamide A, amantamide B	CXCR7 agonist, cytotoxic	Li, T et al. [20]
	*Oscillatoria agardhii* (NIES-205), *Oscillatoria agardhii* (NIES-596)	prenylagaramide A, prenylagaramide B	no biological activity	Murakami, M et al. [21]
**marine animals**				
conch	* Conus imperialis *	conopressin G	vasopressin-like	Nielsen, D.B. et al. [16]
conch	*Conus striatus*	conopressin S	vasopressin-like	Cruz, L.J. et al. [17]
cuttlefish	*Sepia officinalis*	sepiatocin, pro-sepiatocin	muscle contractile activity modulator (sepiatocin)	Henry, J et al. [22]
skate	*Raja porosa*	SP-A, SP-B	antioxidant	Hu, F.Y. et al. [23]

**Table 2 marinedrugs-21-00509-t002:** ^1^H and ^13^C NMR data for xinghamide A (**1**), recorded at 850 MHz and 212.5 MHz, respectively, in DMSO-*d*_6_.

Xinghamide A (1)							
Amino Acids	C/H	δc, Type	δ_H_, Mult (*J* in Hz)	Amino Acids	C/H	δc, Type	δ_H_, Mult (*J* in Hz)
l-Phe	1	172.2, C		l-Pro-2	25	171.4, C	
	2	54.7 CH	3.82, m		26	59.1, CH	4.41, m
	3a	36.5, CH_2_	3.08, dd (13.0, 5.5)		27a	29.0, CH_2_	2.01, m
	3b		3.01, dd (13.0, 4.0)		27b		1.77, m
	4	138.9, C			28a	24.5, CH_2_	1.85, m
	5/9	129.7, CH	7.08, d (7.0)		28b		1.80, m
	6/8	127.4, CH	7.13, d (7.0)		29a	47.3, CH_2_	3.71, m
	7	125.4, CH	7.09, d (7.0)		29b		3.64, m
	2-NH		7.14, d (7.0)	l-Val-3	30	169.4, C	
l-Pro-1	10	169.2, C			31	55.4, CH	4.28, d (8.5)
	11	59.9, CH	4.17, m		32	30.1, CH	1.92, m
	12a	28.5, CH_2_	1.85, dtt (22.0, 12.0, 5.0)		33	19.0, CH_3_	0.87, d (6.5)
	12b		1.77, m		34	18.4, CH_3_	0.83, d (6.5)
	13a	23.8, CH_2_	1.85, m		31-NH		7.81, d (8.5)
	13b		1.50, m	l-Glu	35	170.5, C	
	14a	46.4, CH_2_	3.49, m		36	55.2, CH	4.12, m
	14b		3.37, m		37a	25.0, CH_2_	2.21, m
l-Val-1	15	170.6, C			37b		1.84, m
	16	57.4, CH	4.18, d (8.0)		38a	29.3, CH_2_	2.12, dt (16.5, 8.5)
	17	30.5, CH	1.92, m		38b		2.04, m
	18	19.1, CH_3_	0.78, d (3.0)		39	177.4 CH	
	19	18.1, CH_3_	0.77, d (3.0)		36-NH		7.86, br s
	16-NH		7.79, d (9.5)	l-Pro-3	40	171.2, C	
l-Val-2	20	170.8, C			41	57.4, CH	4.43, dd (8.5, 5.0)
	21	57.8, CH	4.13, m		42a	28.2, CH_2_	2.02, m
	22	30.9, CH	1.92, m		42b		1.88, dd (12.0, 7.0)
	23	19.2, CH_3_	0.81, d (7.0)		43a	24.5, CH_2_	1.94, m
	24	18.2, CH_3_	0.82, d (7.0)		43b		1.76, m
	21-NH		7.93, d (9.0)		44a	47.0, CH_2_	3.72, m
					44b		3.51, m
				l-Thr	45	168.5, C	
					46	56.9, CH	4.38, t (6.5)
					47	66.9, CH	3.83, d (6.5)
					48	19.4, CH_3_	1.11, d (6.5)
					46-NH		8.17, d (7.5)

## Data Availability

Data are contained in the article or Appendix A.

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
