# Peer review of "Xinghamide A, a New Cyclic Nonapeptide Found in Streptomyces xinghaiensis"

_marinedrugs, 2023, doi:10.3390/md21100509_

Round 1

Reviewer 1 Report

The present MS describes the occurrence, isolation and structure determination of a new cyclic peptide. The methodology applied are accurate and falls within the scope of the study. It is understandable that authors might not have any background in structure description but it is important that right expressions are used not to confused readers. As I understood it, the new peptide has been isolated from a natural source. Accordingly, it is confusing to use "produce" in the following "By using a mixture of ethyl acetate and methanol as the solvent, it was 101 possible to produce xinghamide". Likewise, the ion related to the molecule detected in the mass spectrum is the sodium adduct not the protonated ion which in that case would have been [M+H]. Plus, the IR spectrum displays bands of absorption not peaks. 

In the NMR description, I do not see it accurate to define the sequence of amino acids based on the NOE cross peaks as these correlations report the proximity of protons through space. Nothing to see with the scalar interactions exhibited with the other NMR experiments. Two protons can be adjacent but far in space, they will not interact in ROESY. 

In case the interactions presented in Fig 3 and Fig 4 are the same, why using two different types of double arrows for each figure?

As for the stereochemical study, I was expecting a comparison of author's LC-MS data with some references. I cannot understand how the AC was defined for each AA. Authors said "all of the amino acids adhered to the L-configuration" but both L-FDLA and D-FDLA derivatives of the AA are displayed in the SI. Could you provide more information on how the decision was made?

As for the GNPS molecular network, I guess a reader could be interested in knowing the composition of this culture, at least the main groups of chemistry occurring here. Plus, the new compound 1 was part of a cluster, authors did not comment on the other masses of the cluster. How to relate them to the new compound 1? Are they showing any feature relevant enough to be highlighted here?

Author Response

Dear reviewer 1.

We express our gratitude for your valuable feedback on our manuscript. The recommended minor mistakes have been rectified and duly emphasized on the revised manuscript attached. Furthermore, we have included the modifications in a table form. 

Furthermore, the subsequent modifications have been implemented:
1.    In order to enhance the reliability of the sequence established by ROESY, MS/MS data was employed as a supplementary tool. The supplementary figure labeled as S14 is presented.
2.    We have changed the word from ‘adhered’ to ‘determined’ to the section pertaining to the Advanced Marfey response. We also added a reference to Advanced Marfey.
3.    As mentioned in the main text, the reason for including GNPS data in the supporting information was to emphasize that strain YSL1 does not produce a target compound during liquid culture, but only appears during solid culture. Because annotations were not made for other compounds, no additional information was written. Please refer to the text as follow in the manuscript. 'One particular molecular family exhibited distinct characteristics compared to the others detected only in solid culture (Figure S2). This distinction arose from the absence of library matches for the related MS/MS spectra in both native and analog search modes.'
We express my gratitude for your assistance.

Sincerely,

Seung Hyun Kim, PhD. 

Professor, College of Pharmacy, Yonsei University

Reviewer 2 Report

The manuscript reports a structure and an anti-inflammatory activity of a new peptide Xinghamide A from the Streptomyces xinghaienesis isolated from a halophyte species Suaeda maritima. In general, the manuscript is well written, provides sufficient data for structure elucidation of the new compound and determination of its biological activity. However, the impact of this study to natural product biodiscovery or drug discovery is low due to lack of novelty in both chemistry and bioactivity, I would suggest the authors submit this manuscript to another suitable journal.

Author Response

Dear reviewer 2,

While we understand the reviewer's concern about the novelty of our findings, it's essential to emphasize the importance of secondary metabolites in microorganisms living in extreme environments.The isolation and characterization of bioactive compounds from marine organisms, including halophyte-associated microorganisms like Streptomyces xinghaienesis, remain a critical area of research. Our manuscript adds to this body of knowledge by presenting the discovery of Xinghamide A, a new peptide with anti-inflammatory activity, from a unique marine source. Even though the chemical and biological properties of the compound may not be groundbreaking on their own, they still hold potential for further exploration and optimization. In light of these points, we believe that our manuscript still holds value for the scientific community interested in natural product chemistry and drug discovery. We are committed to addressing the other reviewer's comments and improving the manuscript's clarity and presentation. We respectfully request that the reviewer reconsider their recommendation and allow us the opportunity to revise and resubmit our work to this journal. We are confident that with the suggested revisions, our manuscript can make a meaningful contribution to the field.
We express my gratitude for your assistance.

Sincerely 
Seung Hyun Kim, PhD. 

Professor, College of Pharmacy, Yonsei University

Reviewer 3 Report

A new nonapeptide Xinghamide A was obtained from Streptomyces xinghaiensis isolated from a halophyte, Suaeda maritima (L.) Dumort. Based on high-resolution mass and NMR spectroscopic data, the primary structure of A was established with ROESY and HMBC NMR spectra. The absolute configurations of the α-carbon of each amino acid residue were determined with 1-fluoro-2,4-dinitrophenyl-L-and D-leucine amide (Marfey’s reagents) and 2,3,4,6-tetra-O-acetyl-β-D-glucopyranosyl isothiocyanate, followed by LC-MS analysis. The anti-inflammatory activity of xinghamide A was evaluated by inhibitory abilities against the nitric oxide secretion and cyclooxygenase-2 expression in lipopolysaccharide-stimulated RAW264.7 cells.  All scientific evidences support the results. The manuscript was wrote smoothly and logically.  It can be accepted for publication after a minor revision.

Comments

1.  Tandem MS/MS can also be applied in the determination of the peptide sequence. Do the authors attempt? Please see reference:  Rapid Commun. Mass Spectrom. 1996, 10(8), 897–902.

2. English presentation:

 amantamide A and B  -    amantamides A and B

Author Response

Dear reviewer 3,

We express our gratitude for your valuable feedback on our manuscript. We have incorporated tandem mass spectrometry (MS/MS) data to corroborate the structural arrangement of the molecule. Additionally, we have made the adjustment you asked by modifying the nomenclature from 'amantamide A and B' to 'amantamides A and B'.

Sincerely 
Seung Hyun Kim, PhD. 

Professor, College of Pharmacy, Yonsei University

Reviewer 4 Report

In this manuscript, Kim and co-workers carried out a chemical investigation on the Streptomyces xinghaiensis isolated from a halophyte, Suaeda maritima. This study let to the discovery of new cyclic nonapeptide named xinghamide A. Its structure was established by the extensive analysis of NMR, MS and IR spectrum. While its absolute configuration was assigned by advanced Marfey’s method and analysis of GITC derivatives. The structure elucidation was solid. Furthermore, this compound was subjected to anti-inflammatory bioassay, which showed noticeable activity. Based on these findings, this work is important.

However, there are some concerns as following:

1. Figure 1 was too blurry to see the chemical structures clearly.

2. The mass spectrum in Figure S11 was too blurry to see the data clearly.

3. P4L104: Usually, ‘a protonated molecular ion’ was referred to as [M+H]+ not [M+Na]+.

4. As shown in Figure S3, the integral of the chemical shift at δH 7.85 ppm was 1.37, which were not supported there were protons of 36-NH and OH as shown in Table 2. And it seemed a bit suspicious for the chemical shift of OH of COOH at δH 7.85 ppm.

5. P4L114: There were seven methyl carbons as revealed by its chemical structure, not ‘eight methyl carbons’.

6. It seemed there was a cross-peak between H-38b (δH 2.04) of Glu and H-41 (δH 4.43) of Pro-3 in the ROESY spectrum (Figure S10), which was not consistent with the expression ‘the ROESY correlation between H-37 of Glu and H-41 (δH 4.43) of Pro-3 (P5L135)’.

7. The description ‘the β-proton (δH 4.38) of Thr (P5L137)’ was ambiguous. Did ‘β’ refer to the configuration or the position of proton? Indeed, it was one α-proton of Thr as shown in Table 2.

8. P7L169: It's a bit difficult to understand ‘not confirm toxicity another concentration’.

Others:

1. P2L60&L63: ‘amantamide A and B’ → ‘amantamides A and B’

2. P4L123: ‘14-H2 (δH 3.49)’ → ‘H-14a (δH 3.49)’

3. P4L131: ‘H-29 (δH 3.64)’ → ‘H-29b (δH 3.64)’

4. P4L132: ‘H-37 (δH 2.21)’ → ‘H-37a (δH 2.21)’

5. P4L137: ‘H-44 (δH 3.72)’ → ‘H-44a (δH 3.72)

6. Figure 4 caption: Energy-minimized structures’ → ‘The energy-minimized structure

7. P7L169: ‘There for’ → ‘Therefore’

8. P7L181: Delete the parenthesis in the sentence ‘(at a concentration of 6.3...’.

9. P10L283: ‘HRMS2-based’ → ‘HRMS2-based

10. The DOI in the references at the end of this manuscript appeared repatedly.

Author Response

Dear reviewer 4,

We express our gratitude for your valuable feedback on our manuscript. The recommended minor mistakes have been rectified and duly emphasized. Furthermore, we have included the modifications in a table form. 

In relation to No. 4 that you have raised, we concur with your perspective. The HMBC correlation from H-38 to C-39 was reconfirmed and we enlarged them in the supporting information. The HMBC correlation OH-39->C-39 was deleted from the table and figures. To provide further support, we verified the validity of the sequence through MS/MS experiments. (Figure S14) 

Sincerely,
Seung Hyun Kim, Ph D.

Professor, College of Pharmacy, Yonsei University

Round 2

Reviewer 1 Report

I recommend to accept the MS.